# CoCoFL: Communication- and Computation-Aware Federated Learning via Partial NN Freezing and Quantization

**Kilian Pfeiffer**                                    *kilian.pfeiffer@kit.edu*
*Karlsruhe Institute of Technology*

**Martin Rapp**                                       *martin.rapp@kit.edu*
*Karlsruhe Institute of Technology*

**Ramin Khalili**                                    *ramin.khalili@huawei.com*
*Huawei Research Center Munich*

**Jörg Henkel**                                       *henkel@kit.edu*
*Karlsruhe Institute of Technology*

**Reviewed on OpenReview:** *https://openreview.net/forum?id=XJIg4kQbkv*

## Abstract

Devices participating in federated learning (FL) typically have heterogeneous communication, computation, and memory resources. However, in synchronous FL, all devices need to finish training by the same deadline dictated by the server. Our results show that training a smaller subset of the neural network (NN) at constrained devices, i.e., dropping neurons/filters as proposed by state of the art, is inefficient, preventing these devices to make an effective contribution to the model. This causes unfairness w.r.t the achievable accuracies of constrained devices, especially in cases with a skewed distribution of class labels across devices. We present a novel FL technique, *CoCoFL*, which maintains the full NN structure on all devices. To adapt to the devices' heterogeneous resources, CoCoFL freezes and quantizes selected layers, reducing communication, computation, and memory requirements, whereas other layers are still trained in full precision, enabling to reach a high accuracy. Thereby, CoCoFL efficiently utilizes the available resources on devices and allows constrained devices to make a significant contribution to the FL system, preserving fairness among participants (accuracy parity) and significantly improving final accuracy.

## 1 Introduction

Deep learning has achieved impressive results in many domains (He et al., 2016; Huang et al., 2017; Young et al., 2018), and is also being applied in embedded systems such as mobile phones or internet of things (IoT) devices (Dhar et al., 2021). With recent hardware improvements, these devices are not only capable of performing inference of a pre-trained model, but also of on-device training. Hence, federated learning (FL) (McMahan et al., 2017) has emerged as an alternative to central training. An FL system comprises many *devices* that each train a deep neural network (NN) on their private data, and share knowledge by exchanging NN parameters via a *server*. Distributing learning through FL brings many benefits, most importantly preserving the privacy of the end users.

Devices in real-world systems have limited computation, communication, and memory resources for training, varying across devices. For instance, smartphones that participate in an FL system have different performance and memory (e.g., different hardware generations), and the conditions of their wireless communication channels vary (e.g., due to fading (Goldsmith, 2005)). Similar observations can be made in IoT systems (Bhardwaj et al., 2020). As stated by prior art (Rapp et al., 2022; Xu et al., 2021; Diao et al.,

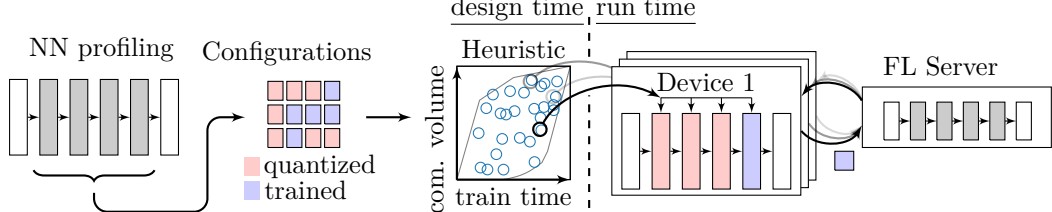

Figure 1: Overview of CoCoFL. At design time, different configurations of frozen/trained layers are profiled w.r.t. communication, computation, and memory in training. At run time, a heuristic selects a suitable configuration on each device w.r.t. the device's constraints.

2020; Horvath et al., 2021), to enable efficient learning in such systems, FL needs to adapt to the per-device constraints, i.e., *hardware-aware FL*. Although different techniques are proposed, the common idea in these state-of-the-art solutions is to reduce the complexity by training subsets of the NN model on less capable devices, to match the required resources for training to the actual resource availability. While these techniques enable constrained devices to participate in the training, they do not effectively learn from their data, i.e., they do not preserve fairness (accuracy parity (Shi et al., 2021)). This is especially critical with non independent and identically distributed (non-iid) data, where the data differs statistically between devices (Hsu et al., 2019). Our evaluation results show that existing solutions perform poorly in non-iid cases, such that in some settings, simply excluding constrained devices from training reaches higher accuracies. We attribute this in part to the fact that updates at constrained devices are less relevant to the overall learning objective, as they train much smaller subsets of the model, and also to the inability of these solutions to efficiently use the available resources in constrained devices.

In this paper, we propose a new technique CoCoFL (Fig. 1), that allows all devices to calculate gradients based on the full model, irrespective of their capabilities, through partial freezing and quantization of the model at constrained devices. We show that *quantizing frozen layers but keeping trained layers at full precision* results in a large reduction in resource requirements, while still enabling efficient learning at devices. This combination has not been exploited so far. Freezing layers reduces the required gradient computations, the storage of intermediate activations, and the size of the parameter update, while quantization further speeds up the computations of frozen layers. Thereby, our solution adjusts the complexity of training to the resources available at each device. Partial freezing and quantization opens up a large design space, where each layer can be frozen or trained on each participating device. The selection of trained layers has a significant impact on the required resources and on the accuracy. We introduce a heuristic that allows for server-independent selection of layers w.r.t. local resource availability at run time, based on design-time profiling of the performance of devices. *We demonstrate that our solution reaches significantly higher accuracy in iid and non-iid data, when compared with the state of the art, significantly improving FL systems.*

In summary, we make the following novel contributions:

- We empirically show that in many scenarios, state-of-the-art subset-based techniques do not reach better accuracies than simply excluding less capable devices (a straightforward baseline). We observe this throughout various datasets (e.g., CIFAR10, XChest, and Leaf benchmark data), data distributions, and NN topologies (e.g., ResNet, DenseNet, and Transformers).
- Compared to the state of the art, in these scenarios, we enable increased fairness of contribution and higher final accuracies in FL with heterogeneous resources by allowing less capable devices to do training based on the full NN structure. This is achieved by the following technical contributions.
- We introduce a novel partial freezing and quantization technique to adjust to computation, communication, and memory constraints of devices that allows to train full layers of NNs.
- We introduce *CoCoFL*[1], based on partial freezing and quantization, with a simple, yet effective, heuristic to select locally on each device which layers to freeze or train based on the available communication, computation, and memory resources.

---

[1]The code is available at `https://github.com/k1l1/CoCoFL`.

## 2    System Model and Problem Definition

**System Model:** We target a distributed system comprising a *server* that is responsible for coordination and *devices* $\mathcal{C}$ that act as clients. Each device $c \in \mathcal{C}$ has exclusive access to its local data $\mathcal{D}_c$. Training is done iteratively using FL in synchronous rounds $r$. In each round, a subset of the devices $\mathcal{C}^{(r)} \subset \mathcal{C}$ is selected. Each selected device downloads the latest model parameters $w^{(r)}$ from the server, performs training on its local data for a pre-defined *round time* $T$, and then uploads the updated model parameters to the server. The server averages all received updates (FedAvg (McMahan et al., 2017)) to build $w^{(r+1)}$ for the next round:

$$w^{(r+1)} = \frac{1}{\sum_{c \in \mathcal{C}^{(r)}} |\mathcal{D}_c|} \cdot \sum_{c \in \mathcal{C}^{(r)}} |\mathcal{D}_c| \cdot w_c^{(r)} \tag{1}$$

The server discards updates that arrive late (straggler), i.e., devices must upload their updates in time.

**Device Model:** Devices are *heterogeneous w.r.t. their computation (performance), memory, and communication constraints.* The performance of a device (how long the training of an NN takes) depends on its hardware (number of cores, microarchitecture, memory bandwidth, etc.), software (employed deep learning libraries, etc.), and training configuration (topology, amount of data, etc.). Similarly, the available memory $M_c$ of device $c$ depends on its hardware, while the required memory during training depends on the software and training configuration. Some of these configurations are fixed by the FL system (NN topology, etc.), while others are fixed by the device (hardware, software, amount of data), but some configuration $A_c^{(r)}$ can be adjusted per device per round. In our case, $A_c^{(r)} \in \mathcal{A}$ describes the subset of all NN layers that are trained in round $r$ by device $c$, with $\mathcal{A}$ being the set of all configurations (see Section 5). The training time of device $c$ for any $A \in \mathcal{A}$ is represented by the function $t_c : \mathcal{A} \to \mathbb{R}$. The required memory during training is represented by $m_c : \mathcal{A} \to \mathbb{R}$. We obtain $t_c$ and $m_c$ through profiling our technique on real hardware (measuring the training time and peak memory usage for different $A$).

The communication channel between devices and servers is commonly asymmetric: The download link from the server to the devices can be neglected due to the commonly high transmit power of base stations (Yang et al., 2020). The upload link from devices to the server is subject to heterogeneous channel quality, as discussed in Section 1. Therefore, we model the communication constraint $S_c^{(r)}$ of a device $c$ in round $r$ as a limit in the number of bits that can be uploaded to the server at the end of the round. In our case, all layers not contained in $A_c^{(r)}$ are frozen (and quantized). Their parameters do not change, hence, do not need to be uploaded to the server. We represent the size of the parameter update for any $A \in \mathcal{A}$ by a function $s : \mathcal{A} \to \mathbb{N}$ that is independent of device characteristics. Function $s$ can be derived analytically or by counting parameters per layer.

**Problem Definition:** Our main objective is to maximize the *final accuracy acc* of the server model $w^{(R)}$ after $R$ rounds under communication, computation, and memory constraints, by selecting per-device per-round the set of trained layers $A_c^{(r)}$:

$$\text{maximize } acc(w^{(R)}) \quad \forall c \in \mathcal{C} \quad \forall 1 \le r \le R \qquad \text{s.t. } t_c(A_c^{(r)}) \le T \ \wedge \ m_c(A_c^{(r)}) \le M_c \ \wedge \ s(A_c^{(r)}) \le S_c^{(r)} \tag{2}$$

We also evaluate fairness (accuracy parity (Shi et al., 2021)) as a secondary metric, by measuring the *device- or group specific accuracy* using data that reflects each device's or group's distribution of local data $\mathcal{D}_c$.

## 3    Related Work

We divide the related work into works that employ a similar mechanism (quantization/freezing) and works that target a similar problem (computation/communication/memory constraints in FL).

**Quantization and Freezing in Centralized Training:** Most works on quantization target the inference, with full-precision training. Naive *training* on quantized parameters leads to training stagnation as small gradients are rounded to zero (Li et al., 2017). To solve this, one branch of works performs stochastic rounding (Gupta et al., 2015). However, stochastic rounding prevents convergence to the local minimum in the final phase of training with a low learning rate (Li et al., 2017), reducing the accuracy. Another

branch of work uses a full-precision copy of the parameters as an accumulator (Micikevicius et al., 2018). Calculating the parameter gradient based on quantized activations and parameters induces instability to the learning processes, which requires a lower learning rate to maintain convergence, slowing down the training, but also still resulting in a lower accuracy (Guo, 2018). *In summary, achieving fast convergence and high accuracy requires keeping the trained layers in full precision (activation and parameters in the forward and backward pass).* Goutam et al. (2020) stochastically freeze layers of an NN to speed up training, keeping the frozen layers at full precision which limits the achievable speedup. Also, due to its stochastic nature, it is not applicable to a hard computation constraint. All these works either apply quantization or freezing. *None of the existing works has exploited the symbiosis between freezing and quantization, where only frozen layers are quantized to maintain good convergence properties.*

**Communication, Computation, and Memory Constraints in FL:** Most works on resource-constrained FL have targeted communication, extensively studying compression, quantization, and sketching of parameter updates (Shi et al., 2020; Thakker et al., 2019). All perform regular training of the full NN in full precision, requiring full computation and memory resources, and only reduce the size of the parameter update. They are orthogonal to ours, i.e., applicable on top of CoCoFL. The work in Chen et al. (2021) detects NN parameters that have stabilized, freezes them, and excludes them from synchronization to reduce the required communication. However, this technique can not cope with a given communication constraint. Recently, a preliminary work (Yang et al., 2022) proposed freezing layers to save communication and memory in FL. It exploits that frozen layers do not require storing activations for computing gradients, and do not need to be uploaded to the server. This technique has later been combined with quantization during download and upload (Ro et al., 2022), but unlike in our CoCoFL, quantization is not used during training, missing out on significant optimization opportunities (e.g., up to $4\times$ lower computation time as we will show in our experiments). All these works do not reduce the computation cost of training.

Computation constraints in FL devices have only recently attracted attention. Some employ asynchronous FL (Xie et al., 2020). However, this does not reduce memory requirements and may reduce the convergence stability (McMahan et al., 2017). FedProx (Li et al., 2020) dynamically drops training data on straggling devices, which reduces computations but does not affect communication and memory requirements, and reduces the contribution of less capable devices. DISTREAL (Rapp et al., 2022) employs dropout to dynamically reduce the size of the trained NN, reducing computations. They still transmit the full NN updates, and, hence, do not reduce communication costs. Several others train subsets of the NN on each device by (temporarily) scaling the width (number of filters/neurons) of layers. This may save communication, computation, and memory. In particular, Helios (Xu et al., 2021), HeteroFL (Diao et al., 2020), and FjORD (Horvath et al., 2021) proposed to create a separate subset per each device according to its available resources. However, training very small subsets on weak devices does not enable them to effectively learn from their data, as we show in our evaluation, reducing fairness, hence, the final accuracy. Additionally, the reductions in communication and computation achieved by width scaling are tightly coupled. Consequently, one of them forms the bottleneck, resulting in unexploited resources in the other metric. Finally, Yang et al. (2020) study the trade-off between communication and computation to minimize the overall energy consumption. This work is not applicable to a per-device computation or communication constraint.

*In summary, none of the existing works on resource-constrained FL can adapt to per-device communication, computation, and memory constraints, while still effectively learning from all data on all devices. We achieve this through our novel combination of freezing and quantization.*

## 4 Partial Freezing and Quantization

This section introduces our freezing, layer fusion, and quantization technique to reduce the training complexity, which will be used in the CoCoFL algorithm (Section 5).

### 4.1 Background: NN Structure and Training

**Structure:** Common state-of-the-art deep NNs like ResNet (He et al., 2016), DenseNet (Huang et al., 2017), or MobileNet (Howard et al., 2017) follow a similar structure, where a convolutional layer (CL) is followed

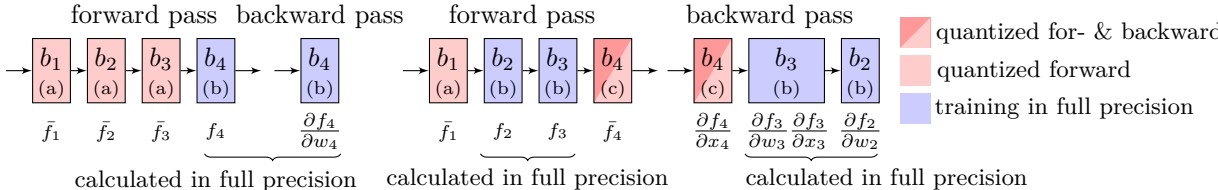

Figure 2: The left part shows an NN where only block $b_4$ is trained, while all others are frozen, and the forward passes of $b_1, b_2,$ and $b_3$ are quantized ($\bar{f}_1, \bar{f}_2,$ and $\bar{f}_3$), not requiring a backward pass. The right part shows two blocks $b_2, b_3$ being trained, therefore requiring intermediate gradients of the quantized block $b_4$. Blocks are labeled (a), (b), and (c) depending on their types.

by a batch normalization (BN) layer, followed by ReLU activation. These layers account for the majority of training time. We label repeating structures like this a *block*, where in a general case, an NN comprises $N$ blocks. We treat blocks as the smallest entity that is either frozen or trained. Note that our technique is also applicable to variants of NNs, (e.g., multiple skip connections or Transformers as we show in our evaluation) but for the sake of simplicity, the block description follows the ResNet structure.

**Training:** An update step in NNs comprises a forward and a backward pass. We describe the forward pass as a chain of consecutive operations, where each block $i$ has an associated forward function $x_{i+1} = f_i(x_i)$ with parameters $w_i$. The full forward pass of the NN is calculated as $\hat{y} = f_N(f_{N-1}(\ldots(f_1(x_1))))$, where $\hat{y}$ is the NN's output and $x_1$ is the input. The backward pass consists of several gradient calculations to compute the parameter gradients $\frac{\partial E}{\partial w_i}$ for each block $i$, where $E = \mathcal{L}(\hat{y}, y)$ is the optimization criterion with some loss function $\mathcal{L}$. Using the chain rule, the calculations can be split into several gradient calculations. In the general case, the gradients w.r.t. a block's parameters can be expressed as

$$\frac{\partial E}{\partial w_i} = \frac{\partial E}{\partial \hat{y}} \Big( \prod_{k=i+1}^{N} \frac{\partial f_k(x_k)}{\partial x_k} \Big) \frac{\partial f_i(x_i)}{\partial w_i}. \tag{3}$$

Using the calculated gradients $\frac{\partial E}{\partial w_i}$, local training with stochastic gradient descent (SGD) obtains updated parameters $\tilde{w}_i = w_i - \eta \frac{\partial E}{\partial w_i}$, where $\eta$ is the learning rate. When calculating gradients of several blocks, intermediate gradient computations can be reused.

### 4.2 Freezing, Fusion, and Quantization of Blocks

**Freezing:** Freezing a parameter removes the need to calculate its gradients. As by Eq. (3), the number of required intermediate gradients depends on the block's index (e.g., the calculation of $\frac{\partial E}{\partial w_N}$ requires no intermediate gradients, while $\frac{\partial E}{\partial w_1}$ requires intermediate gradients from all other blocks). Based on the required per-block operations, we distinguish between three block types (illustrated in Fig. 2):

(a) *Frozen block:* With no preceding trained block, a frozen block only requires a forward pass $f_i(x)$.
(b) *Trained block:* Trained blocks require a forward pass $f_i(x_i)$, calculation of gradients w.r.t. their parameters $\frac{f_i(x_i)}{\partial w_i}$, and gradients w.r.t the input $\frac{\partial f_i(x_i)}{\partial x_i}$ for preceding trained blocks.
(c) *Frozen block with backward pass:* With preceding trained blocks, a frozen block requires the forward pass $f_i(x_i)$ and intermediate gradients w.r.t. the input $\frac{\partial f_i(x_i)}{\partial x_i}$.

*Consequently, freezing blocks reduces the number of per-block operations of frozen blocks (from 3 to 2 or 1), and therefore saves multiply-accumulate operations (MACs), reducing computation time.* Similarly, if a layer is not trained, the activation values $x_i$ can be released in memory during the forward pass, reducing the memory footprint. Additionally, parameters of frozen layers do not change throughout an FL round, therefore, do not have to be uploaded.

**Fusion:** If BN is used for normalization, we fuse the convolution operation with the following BN operation (Ioffe & Szegedy, 2015; Jacob et al., 2018) in frozen layers (we study the application of other normalization techniques in Section 6). A BN layer normalizes each channel to zero mean and unit variance followed by a trainable scale $\gamma_i$ and bias $\beta_i$. The statistics of frozen blocks that do not require intermediate gradients

(type (a)) stay constant over time. Hence, we can express the BN operation as a linear operator with $y_{i_{\mathrm{BN}}}$ being the channel-wise output of the BN operation, $y_{i_{\mathrm{CL}}}$ the CL's output, and $\epsilon$ a small number for stability

$$y_{i_{\mathrm{BN}}} = \frac{\gamma_i}{\sqrt{\sigma_i^2 + \epsilon}} \cdot y_{i_{\mathrm{CL}}} + \left( \frac{-\mu_i \gamma_i}{\sqrt{\sigma_i^2 + \epsilon}} + \beta_i \right), \tag{4}$$

where the coefficient of $y_{i_{\mathrm{CL}}}$ is a new combined scale, referred to as $\hat{\gamma}_i$, and the second summation term is a new combined bias, referred to as $\hat{\beta}_i$. To fuse the BN operation with the preceding CL, we express the CL as $y_{i_{\mathrm{CL}}} = \boldsymbol{W}_i \cdot x$, and plug the output $y_{i_{\mathrm{CL}}}$ into the BN operator: $y_{i_{\mathrm{BN}}} = (\hat{\gamma}_i \boldsymbol{W}_i) \cdot x + \hat{\beta}_i$. This gives a scaled version of the original kernel $\hat{\boldsymbol{W}}_i = \hat{\gamma}_i \boldsymbol{W}_i$ with a new bias $\hat{\beta}_i$. The same can be applied for type (c) layers, with the only difference being that $\mu$ and $\sigma^2$ values are only valid for a limited number of mini-batches. *In summary, the forward pass of type (a) and type (c) blocks is simplified by fusing three operators, reducing the number of operations.*

**Quantization:** Quantization of operators in NNs is usually used for inference. In contrast, we apply the idea for training; however, we **quantize only parts of the NN that are frozen.** Note that this is different from *quantization aware training*, since all trained parameters remain in full precision. Therefore, type (a) and (c) blocks are quantized, i.e., the fused convolution is performed in `int8` instead of `float32`. Additionally, in type (c) blocks, we also quantize the calculation of the intermediate gradients in the backward pass, i.e., the fused transposed convolution. *Consequently, the remaining operations in the forward and backward pass of frozen blocks require less time for execution and have a lower memory footprint.*

Quantization of operations introduces quantization noise in the forward pass and the backward pass of frozen layers, thereby affecting the training. Additionally, updating the fused layers' statistics only at the beginning of the round introduces an error. We demonstrate in our experiments that the benefits of increased efficiency w.r.t. training time and memory footprint outweigh the added noise. We quantify the benefits and the effects of quantization noise in detail in an ablation study in Section 6.3. *In summary, freezing, fusion, and quantization of selected blocks lower the computational complexity, while still allowing less capable devices to calculate parameter gradients of other blocks in full precision based on the full NN. This novel combination has not yet been exploited for training.*

### 4.3 Implementation in PyTorch

We implement the presented training scheme in PyTorch 1.10 (Paszke et al., 2019), which supports `int8` quantization. While the following description could also be applied to other quantization schemes (e.g., `int4`, `float16`), as of now, PyTorch only provides the necessary operators in the backends `FBGEMM` and `QNNPACK` for `int8`. Quantization levels, such as `int4`, could further lower the training time but at the same time could also have an impact on the accuracy. Contrary to quantization for inference, the layers' input scales, as well as the BN layers' statistics change throughout the training. Our implementation enables real-world training time and memory reduction through a combination of on-the-fly scale calculation and statistics from the server.

**Quantization in the Forward Pass:** To preserve a high accuracy, the quantization of a PyTorch tensor $x$ requires a scale $s_x$ to optimally utilize the `int8` range. We calculate the scale by using

$$s_x = s(x) = 2 \cdot \max(|\max(x)|, |\min(x)|)/127.0. \tag{5}$$

Quantized operators (e.g., `linear`, `conv2d`, and `add`) take a quantized tensor $\bar{x}$ (and quantized weights) as input. In the used backends calculations are done in `int8` arithmetic, but the accumulation of the result is done in `int16/32`. Therefore, each quantized operator requires an output scale $s_o$ to map the `int16/32` output to `int8`. This output scale $s_o$ is calculated using Eq. (5). For blocks of type (a), for every mini-batch, we calculate the scale of the input tensor $s_x$ on the fly at the beginning of the first type (a) block. The input gets quantized and stays in the quantized representation throughout its forward propagation through type (a) blocks. For input scales $s_x$, the scale calculation results in negligible overhead, as $x$ is already available in its `float32` representation. However, the output scales $s_o$ depend on the output of an operation (`linear`, `conv2d`, `add`), therefore, can not be calculated a priori without performing the full operation in float. Because of this PyTorch limitation, the output scale $s_o$ and the BN layer's statistics ($\sigma^2$ and $\mu$ from Eq. (4)) are obtained from the server and are only set once per FL round.

**Quantization in the Backward Pass:** Out of the box, PyTorch's *Autograd* system does not support a quantized backward pass but expects `float32` values for each calculated gradient. We implement a custom PyTorch *Module* for blocks of type (c) based on a custom *Autograd Function* that encapsulates all quantized operations in one *backward* call. Due to this limitation, a quantization/dequantization is required, for each block each mini-batch. The intermediate gradients' scale $s_g$ is calculated on the fly. These limitations are inherently considered in our experimental results (profiling), i.e., fixing these limitations of PyTorch would further increase the efficacy of CoCoFL. These overheads are minor compared to the speedups gained through quantization, since large convolution operations dominate the training time (e.g., we measure a 6% overhead of scale/quantization/dequantization of type (c) blocks in MobileNet (Howard et al., 2017) but gain a reduced computation time by a factor of 1.3). Similar to type (a) blocks, the output scales of operators (e.g., transposed convolution) in the backward pass are obtained from the server.

Type (b) blocks (trained blocks) require only minor modifications. In order to acquire the scale of the operations, PyTorch forward and backward `hooks` are used to calculate $s_o$. For trained blocks, $s_o$ can be efficiently acquired since trained blocks' operators calculate regular `float32` outputs. Together with the trained parameters, these scales are uploaded to the server and averaged alongside the NN parameters. The scales only have to be calculated in the last mini-batch of a training round and result in negligible overhead. To perform fusion, as presented in Section 4, devices that train a respective block upload their BN statistics. The statistics are averaged alongside the parameters and distributed to the devices that require the respective statistics for fusion.

**Transformer-Specific Implementation Details:** We treat *encoder* layers as blocks, and quantize linear, layernorm, and ReLU operations for type (a) blocks. Due to PyTorch limitations, the attention mechanism has to remain in `float32`. In type (c) blocks, linear layers and their intermediate gradient calculations get quantized. Further details are provided in Appendix B.

## 5    Overall CoCoFL Algorithm

Partial freezing and quantization enables to adjust the required communication, computation, and memory resources by selecting which blocks to train or freeze. We present CoCoFL, which enables each device to select the trained/frozen blocks based on its available resources, and the required changes in aggregation at the server, in order to maximize the accuracy under constrained resources.

**Heuristic Configuration Selection**: A selection of trained blocks is a training *configuration $A \in \mathcal{A}$*. The set of all configurations of an NN with $N$ blocks comprises $|\mathcal{A}| = 2^N$ configurations (each block is either trained or frozen). In each round, each device can select a separate configuration. Therefore, the total search space in each round is $2^{N \cdot |\mathcal{C}|}$, which is infeasible to explore in its entirety, and impractical as it depends on the parameters like the NN structure. Simplifying the search space by assigning a separate quality measure to each configuration also does not work, since the accuracy after training with a certain configuration depends also on the configurations used by other devices. Therefore, heuristic optimization is required. Simple deterministic heuristics like selecting configurations that train the maximum number of blocks at once show bad performance, as some blocks would never get trained. As another example, a round-robin selection would lead to all devices selecting the same configuration, where our observations have shown that this leads to lower accuracy (we provide experimental results in Appendix C). Therefore, CoCoFL selects a random configuration on each device based on its available resources. Thereby, the probability that many devices select the same configuration is negligible, while eventually all blocks within a device's capability get trained. An additional benefit of this scheme is that no signaling between the server and devices to transmit the available resources and selected configurations is required, which otherwise could slow down the overall FL process and prevent scalability.

To be able to select configurations w.r.t. the available resources, we need to quantify the resource requirements per configuration. We obtain this information through design-time profiling of a real implementation of our presented freezing and quantization scheme on real devices, but an analytical model of the resources could also be employed. Profiling takes several seconds per configuration. For instance, profiling MobileNet takes 4.7 s on average on the x64 target platform. It is, therefore, infeasible to profile all $2^N$ configurations per device, which would take several months. We solve this by only considering configurations $\hat{\mathcal{A}} \subseteq \mathcal{A}$ that

---

**Algorithm 1** Each Selected Device $c$ (Client) in Each Round

---

**Require:** $s, t_c, m_c, \mathcal{D}_c, T, S_c, M_c$           ▷ profiling information of the device, data, available resources
    receive $w^{(r)}$ from the server            ▷ initial parameters at the beginning of the round
    $\mathcal{A}_f \leftarrow \{A \in \hat{\mathcal{A}} : t_c(A) \le T \ \wedge \ m_c(A) \le M_c \ \wedge \ s(A) \le S_c\}$       ▷ feasible configurations (Eq. 2)
    $\mathcal{A}_{\max} \leftarrow \{A_j \in \mathcal{A}_f : \forall A_k \in \mathcal{A}_f : A_j \not\subset A_k\}$        ▷ discard non-maximal configurations
    $A_c^{(r)} \leftarrow$ random choice($\mathcal{A}_{\max}$)            ▷ select random configuration
    $(w_{\text{train}}, w_{\text{quant}}) \leftarrow$ apply($A_c^{(r)}, w^{(r)}$)       ▷ apply the configuration (freeze/quantize blocks)
    $\tilde{w}_{\text{train}} \leftarrow$ train model $(w_{\text{train}}, w_{\text{quant}})$ with local data $\mathcal{D}_c$         ▷ local training
    send $\tilde{w}_{\text{train}}$ to the server            ▷ parameter update

---

**Algorithm 2** FL Server (Synchronization and Aggregation)

---

    $w^{(1)} \leftarrow$ random initialization
    **for each** round $r = 1, 2, \ldots, R$ **do**
       $\mathcal{C}^{(r)} \leftarrow$ select devices
       broadcast $w^{(r)}$ to selected devices $\mathcal{C}^{(r)}$
       **for each** $c \in \mathcal{C}^{(r)}$ **do** receive $\tilde{w}_{\text{train},c}$ from device $c$
       **for each** block $i$ **do**            ▷ aggregation
          $\mathcal{C}_i \leftarrow \{c : \tilde{w}_{\text{train},c} \text{ contains block } i\}$      ▷ devices that have trained block $i$
          $w_i^{(r+1)} \leftarrow \left(1 - \frac{|\mathcal{C}_i|}{|\mathcal{C}^{(r)}|}\right) \cdot w_i^{(r)} + \frac{1}{|\mathcal{C}^{(r)}|} \sum_{c \in \mathcal{C}_i} \tilde{w}_{\text{train},c}(i)$

---

train a single contiguous range of blocks. This reduces the search space to $|\hat{\mathcal{A}}| = \frac{N(N+1)}{2}$, i.e., $17\,\text{min}$ for MobileNet on x64. If resources can be estimated much faster, relaxing this restriction could further improve our technique. The run-time algorithm for devices is outlined in Algorithm 1. At the beginning of each round, each device determines the set of feasible configurations $\mathcal{A}_f \subseteq \hat{\mathcal{A}}$, given its currently available resources. We then discard all configurations that train a subset of blocks trained by another feasible configuration, i.e., we only keep maximal configurations $\mathcal{A}_{\max} \subseteq \mathcal{A}_f$, thereby maximizing the accuracy by fully exploiting the available resources. Each device selects a random remaining configuration, which results in different configurations being trained on different devices without requiring any synchronization between devices. Finally, the selected fusion and quantization configuration is applied, and the NN is trained.

**Aggregation of Partial Updates**: Each device $c$ only uploads updates of the blocks that were trained in full precision ($\tilde{w}_{\text{train},c}$), hence, not frozen or quantized. The server (Algorithm 2) weighs the updates based on the number of devices that have trained each block to account for partial training on the devices. Fig. 1 shows CoCoFL in a nutshell.

## 6 Experimental Evaluation

Partial quantization of NN models results in hardware-specific gains in execution time and memory. Hence, our evaluation follows a hybrid approach, where we profile on-device training loops on real hardware and take the profiling information to perform simulations of distributed systems. This allows for the evaluation of large systems with hundreds or thousands of devices.

### 6.1 Evaluation Setup

**Profiling Setup and Results:** We employ two different hardware platforms to factor out potential micro-architecture-dependent peculiarities w.r.t. quantization or freezing: x64 AMD Ryzen 7 and a Raspberry Pi with an ARMv8 CPU. For each configuration in $\hat{\mathcal{A}}$, we measure the execution time, maximum memory usage, and upload volume. The measurements are stored in a lookup table for the FL simulations. NN-specific details about $N$, $|\hat{\mathcal{A}}|$, and the used platform are given in Table 1, further details in Appendix A. For simplicity in the implementation, we select one *skip connection block* in ResNet, MobileNet, and DenseNet as the smallest entity that is either trained or frozen. This choice allows for limited implementation overhead, as the structure is repeatedly used in the NNs. The block granularity could be further increased by selective training and freezing of CL layers within a skip connection block, but it would require more individual cases to be implemented. Exemplary profiling results of MobileNet on x64 are shown in Fig. 3, where all quantities

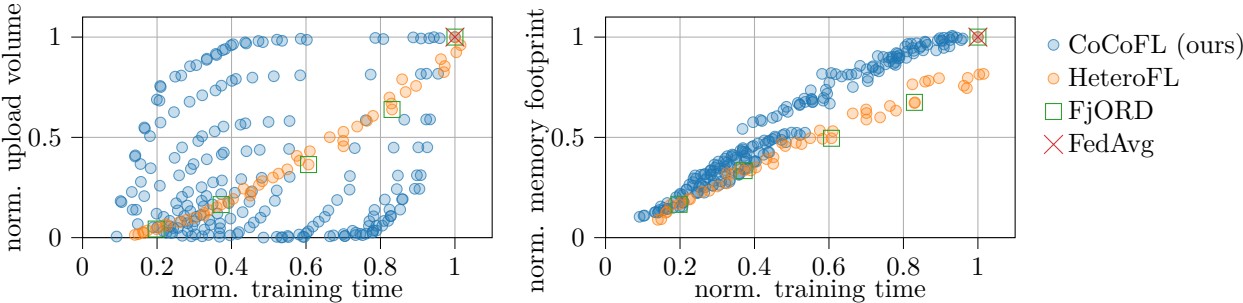

Figure 3: Profiling of MobileNet regarding upload volume, training time, and memory footprint, normalized to training the full NN with FedAvg. Each marker represents a training configuration. The figure shows that CoCoFL supports the same range w.r.t. to the constraints as HeteroFL and FjORD but enables independent adjustability of computation/memory and communication.

Table 1: Hyperparameters of FL experiments.

| Hyperparameters | DenseNet CIFAR10/ (CINIC10) | MobileNet CIFAR10 | MobileNet CIFAR10 GroupNorm | ResNet50 CIFAR100 | ResNet18 FEMNIST | MobileNet XChest | Transformer IMDB/ (Shakespeare) |
|---|---|---|---|---|---|---|---|
| Rounds $R$ | 800 | 1000 | 1000 | 800 | 600 | 1000 | 1000 |
| Amt. data $|\mathcal{D}|$ | 50K (90K) | 50K | 50K | 50K | 640.5K | 12.7K | 40K (200K) |
| $|\mathcal{C}^{(r)}|/|\mathcal{C}|$ | 10/100 | 10/100 | 10/100 | 10/100 | 35/3500 | 10/100 | 10/100 |
| $\eta$-decay ($\times 0.1$) | [750] | [600, 800] | [800] | [750] | [400] | [600] | [800] |
| Weight Decay | 0.001 | 0.01 | 0.001 | 0.01 | 0.01 | 0.01 | - |
| Nb. configs $|\hat{\mathcal{A}}|$ | 253 | 210 | 210 | 171 | 55 | 210 | 35 |
| Nb. blocks $N$ | 23 | 21 | 21 | 19 | 11 | 21 | 8 |
| Platform | ARM | x64 | x64 | x64 | ARM | x64 | x64 |

are normalized to FedAvg (full training of the NN). Using the same setup, we profile training with subsets of the NN's filters, as employed by HeteroFL and FjORD. In CoCoFL, a configuration refers to a specific selection of blocks that are frozen, quantized, and fused. In HeteroFL/FjORD, a configuration refers to a specific ratio of filters that are trained and filters that are dropped. As a consequence, the trade-offs vary. Our results show that the combination of freezing, fusion, and quantization allows training with configurations that reduce the execution time by up to 90% and the memory footprint by 89% compared to full training of the NN, a similar range as HeteroFL and FjORD. Further, CoCoFL enables independent adjustability of computation/memory and communication, giving us a higher degree of freedom to select a configuration that utilizes the resources at the device, therefore, more efficiently utilizing available resources. Contrary to that, training subsets results in a tightly coupled reduction of resources. However, in some cases, CoCoFL is required to pick a non-optimal configuration w.r.t. computation, to satisfy a memory constraint.

**FL Setup and Hyperparameters:** We evaluate our technique in an FL system, using the profiling results. For each experiment, we distribute the data from the datasets CIFAR10/100 (Krizhevsky & Hinton, 2009), FEMNIST (Cohen et al., 2017), CINIC10 (Darlow et al., 2018), XChest (Wang et al., 2017), IMDB (Maas et al., 2011), and Shakespeare (Caldas et al., 2019) to devices in $\mathcal{C}$. We evaluate ResNet (Gao et al., 2020), DenseNet (Huang et al., 2017), MobileNet (Howard et al., 2017), and Transformer (Vaswani et al., 2017) NN models. In each round, a subset $\mathcal{C}^{(r)}$ is selected for participation. Devices are randomly grouped in three equally sized sets. The set of *strong* devices is capable of training the full NN and uploading all parameters, with no memory constraints. The round time $T$ is set to the time a *strong* device requires to finish one training round. The set of *medium* devices has 2/3 of the computational and memory resources of the *strong* devices. Hence, to match the round time $T$, the set of *medium* devices has to select configurations that reduce required computations to 2/3 of *strong* devices. The set of *weak* devices has 1/3 of the computation and memory capabilities of the *strong* devices. We model the communication budgets of *medium* and *weak* devices randomly over rounds to simulate an environment with varying communication channel quality, s.t. $S_{c_{\text{medium, weak}}}^{(r)} \sim \mathcal{U}(\frac{S_{\text{strong}}}{2}, S_{\text{strong}})$. We compare CoCoFL to several baselines: state-of-the-art HeteroFL

Table 2: Accuracy (Top 1) in % for DenseNet, MobileNet, ResNet18, ResNet50, and Transformer. For XChest the F1 macro score is given (unbalanced data). In almost all scenarios CoCoFL outperforms the baselines, reaching higher final accuracies.

| Topology | DenseNet | | | MobileNet | | | | | ResNet18 | |
|---|---|---|---|---|---|---|---|---|---|---|
| Setting | CIFAR10 | | | CIFAR10 | | CIFAR10 (w. GroupNorm) | | | FEMNIST | |
| Dirichlet $\alpha$ | − (iid) | n.-iid@0.1 | rc@0.1 | − (iid) | rc@0.1 | - (iid) | n.-iid@0.1 | rc@0.1 | − (iid) | rc@0.1 |
| Centralized | | 87.8±0.2 | | | 87.0±0.7 | | 82.5±1.1 | | | 88.1±0.0 |
| FedAvg (f. res.) | 84.3±0.1 | 75.6±1.5 | 74.9±3.3 | 84.9±0.2 | 77.4±2.7 | 76.1 ± 0.1 | 69.4 ± 0.5 | 72.9 ± 1.4 | 86.2±0.1 | 82.9±1.2 |
| CoCoFL (ours) | **82.0±0.2** | **71.9±1.8** | **68.8±4.6** | **83.2±0.3** | **72.4±2.9** | **71.3 ± 0.1** | **61.2 ± 1.4** | **63.6 ± 3.5** | 85.0±0.1 | **81.5±0.6** |
| FjORD | 73.7±0.1 | 60.4±2.1 | 48.8±6.8 | 79.1±0.3 | 51.9±7.3 | 64.4 ± 0.6 | 42.9 ± 1.7 | 47.0 ± 5.0 | 85.5±0.0 | 69.3±8.3 |
| HeteroFL | 76.4±0.3 | 64.0±2.4 | 51.2±7.4 | 79.5±0.2 | 53.0±7.6 | 64.8 ± 0.1 | 55.2 ± 0.3 | 47.5 ± 5.0 | 85.9±0.0 | 70.9±5.8 |
| FedAvg | 76.5±0.1 | 60.4±4.2 | 50.9±7.5 | 78.1±0.4 | 49.9±8.7 | 56.2 ± 0.8 | 52.8 ± 1.1 | 45.9 ± 4.7 | **86.1±0.1** | 64.9±7.8 |

| Topology | ResNet50 | | DenseNet | | | MobileNet(large) | | | TF | TF-S2S | |
|---|---|---|---|---|---|---|---|---|---|---|---|
| Setting | CIFAR100 | | CINIC10 | | | XChest | | | IMDB | Shakespeare | |
| Dirichlet $\alpha$ | − (iid) | rc@0.1 | − (iid) | n.-iid@0.1 | rc@0.1 | − (iid) | n.-iid@0.5 | rc@0.5 | − (iid) | − (iid) | −rc (Leaf) |
| Centralized | 61.6±0.4 | | | 80.5±0.2 | | | 94.2±0.2 | | 84.7±0.7 | 52.9±0.7 | |
| FedAvg (f. res.) | 57.0±0.3 | 53.0±0.6 | 77.2±0.1 | 53.9±2.3 | 65.1±1.1 | 94.1±0.3 | 85.9±1.8 | 93.2±0.2 | 82.6±0.4 | 49.1±0.1 | 49.4±0.1 |
| CoCoFL (ours) | **52.5±0.2** | **41.8±2.5** | **73.6±0.1** | **53.5±4.3** | **52.4±7.3** | **91.3±0.3** | **73.0±6.4** | **87.3±3.8** | **82.5±0.5** | **49.3±0.3** | **49.1±0.3** |
| FjORD | 43.6±0.8 | 29.6±4.3 | 65.1±0.7 | 49.2±2.2 | 41.1±6.9 | 66.3±0.9 | 52.7±3.9 | 62.4±0.8 | 78.5±0.7[1] | 42.9±0.5 | 43.0±0.3 |
| HeteroFL | 45.9±0.7 | 31.0±2.3 | 69.4±0.2 | 50.4±2.4 | 43.4±7.2 | 69.4±1.0 | 65.0±0.9 | 65.4±1.6 | 79.2±0.3[1] | 44.1±0.2 | 44.1±0.2 |
| FedAvg | 35.2±0.2 | 23.7±0.4 | 67.7±0.4 | 48.3±2.5 | 42.4±7.2 | 68.2±1.0 | 67.0±0.6 | 66.8±1.4 | 78.5±0.6 | 40.5±0.3 | 40.3±0.1 |

[1]No configuration for *weak* devices available, therefore *weak* devices are dropped.

and FjORD, which are the closest to our technique, as both allow for a per-device reduction of computational resources, as well as upload volume, and memory. Additionally, we compare to a theoretical bound and a straightforward baseline that drops all but the *strong* devices from FL training.

- Centralized: All data is centralized (on one device), serves as a theoretical upper bound.
- FedAvg (full resources) (McMahan et al., 2017): FL is applied, but all devices have full (homogeneous) resources, hence, FedAvg has only one configuration, that is training the full network. This baseline serves as a theoretical upper bound.
- HeteroFL (Diao et al., 2020): A FedAvg variant that drops a number of the CL filters, defined by a shrinkage ratio, where each ratio represents a training configuration. We set it to the maximum that a device can train.
- FjORD (Horvath et al., 2021): Similarly, state-of-the-art FjORD drops CL filters to reduce resources. In difference to HeteroFL, each device trains different drop levels (within their capabilities). Devices switch each mini-batch between a feasible configuration. The paper proposes to use $[20, 40, 60, 80, 100]\%$ of the filters, which results in 5 configurations.
- FedAvg (McMahan et al., 2017): Devices that can not train the NN (e.g., due to limited memory) are dropped from the training (therefore also their data). The reduced set of devices performs FedAvg. This serves as a naive baseline and is known to be used in production use cases (Yang et al., 2018).

We train with the optimizer SGD with an initial learning rate $\eta$ of 0.1. For a fair comparison we do not use momentum, as FjORD is incompatible with a stateful optimizer. The remaining NN-specific hyperparameters, learning rate decay, and weight decay are given in Table 1. For each FL experiment, we report the average accuracy and standard deviation after $R$ rounds of training using 3 independent seeds. We study several data split scenarios: First, an iid case, where data is randomly distributed to all devices, and hence, every device has about the same number of samples per class. Second, we study a non-iid case, where we vary the non-iid-ness with the value of $\alpha$ of a Dirichlet distribution, similar to Hsu et al. (2019). Hereby, the number of samples per class varies between devices. Thirdly, we consider a scenario where data is *resource correlated non-iid* (*rc-non-iid*). This means that information about certain classes is only available on specific device groups, increasing the necessity to include them in the FL process (Fig. 4a). This has recently been identified as a relevant use case in real-world deployments (Maeng et al., 2022). Similarly, the rate of rc-non-iid-ness is controlled with $\alpha$.

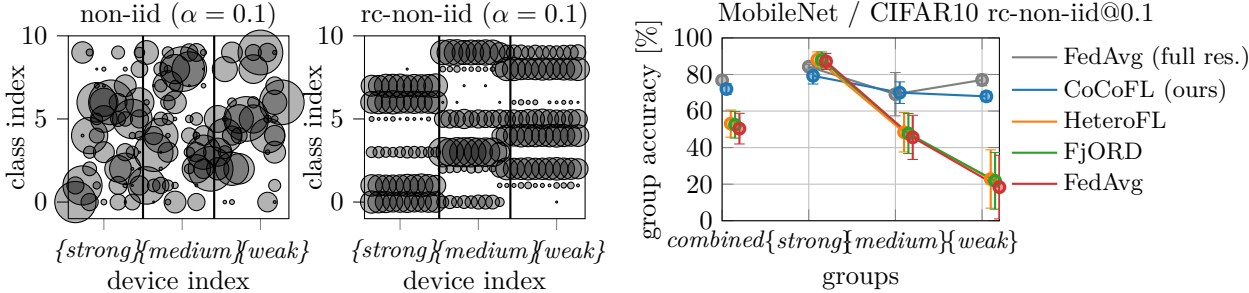

(a) Demonstration of non-iid data (left) and resource-correlated-non-iid data (right). The size of the circles represents the number of samples.

(b) Group accuracies of MobileNet with CIFAR10 and rc-non-iid@0.1 data. CoCoFL preserves fairness among groups, since *weak* and *medium* devices achieve similar group accuracies as if they would have full resources.

Figure 4: Fig. 4a visualizes resource-correlated non-iid, while Fig. 4b shows the effects of this distribution w.r.t. the fairness (group accuracy) in CoCoFL and baselines.

## 6.2 FL Results

Table 2 presents the accuracy results given in %. For XChest (unbalanced) the F1 macro score is given.

**Vision Models:** For iid data, CoCoFL performs close to FedAvg (full res.), improving the final accuracy over the baselines by 5.5 p.p. for DenseNet (CIFAR10), by 3.7 p.p. for MobileNet (CIFAR10), and by 6.6 p.p. for ResNet50 (CIFAR100). Similar trends can be seen for CINIC10. This clearly indicates that CoCoFL uses the available resources on devices more effectively. An outlier is the FEMNIST dataset, where FedAvg reaches the highest accuracy despite dropping 2/3 of the devices. We attribute this to the high number of redundant samples in the dataset (10K per class compared to with 5K and 500). Contrary to that, if the number of samples is more limited, as it is the case in the XChest experiments (12K samples total), the advantage of CoCoFL over the baselines increases.

The necessity to include less capable devices in the FL training is more clearly visible in cases with rc-non-iid. In Table 2, it can be seen that $\alpha = 0.1$ results in a larger gap between the upper bound and the naive baseline, demonstrating the importance of involving all devices in the training. CoCoFL enables *weak* and *medium* devices to contribute to the global model, reaching up to 20 p.p. higher accuracy compared with the state of the art. The reason is that CoCoFL allows *weak* and *medium* devices to calculate gradients based on the full NN. In the case of DenseNet and MobileNet, FjORD and HeteroFL even perform close or inferior to the naive baseline, which excludes *weak* and *medium* devices from the training, failing to preserve fairness (accuracy parity) among devices. Similar conclusions can be driven from ResNet18/50, albeit FjORD and HeteroFL perform a bit better in these settings compared with FedAvg. For XChest we present results with $\alpha = 0.5$ since we observe that 0.1 leads to a complete separation of the binary labels, causing all algorithms to fail to learn at all.

**Fairness in Rc-non-iid Scenarios:** To quantify the contribution *medium* and *weak* devices make in the training, we calculate the device-specific accuracy per group (*group* accuracy), where the class accuracies are weighted by the groups' class densities. As it can be seen for MobileNet (rc-non-iid@0.1) in Fig. 4b, CoCoFL achieves group accuracies of 79%/70%/68% for *strong*, *medium*, and *weak* devices, hence, close to the accuracy parity of FedAvg (with full resources). The baselines HeteroFL and FjORD reach 88%/48%/23%, meaning less capable devices can not make a meaningful contribution to the global model, hence, lowering the fairness among the device groups.

**Other Normalization Techniques**: We replace BN with GroupNorm (Wu & He, 2018) in MobileNet to test the robustness of CoCoFL w.r.t. other normalization techniques in vision tasks. For MobileNet with CIFAR10, we observe that the overall accuracy is lower in all evaluated algorithms (Table 2). However, the general trends are similar to MobileNet with BN, i.e., independent from the normalization, CoCoFL outperforms the state of the art. Additionally, we evaluate the rc-non-iid scenario, where CoCoFL reaches

group accuracies of $63.7\pm3.5\%$, $60.7\pm5.6\%$, and $58.3\pm9.8\%$ for *strong*, *medium*, and *weak* devices, whereas FjORD and HeteroFL reach $80.7\pm7.2\%/42.8\pm8.5\%/18.3\pm14.4\%$ and $81.3\pm7.1\%/42.8\pm9.0\%/19.0\pm13.5\%$, respectively. Thus, CoCoFL provides better fairness independent of the normalization technique.

**NLP Models**: To show the applicability of CoCoFL to natural language processing (NLP) problems, we adapt our freezing and quantization scheme (Section 4) for Transformer. We study text classification with the IMDB dataset and next character prediction with the Shakespeare dataset. The Transformer model uses 6 *encoder* layers, with embedding size of 128, hidden size of 128, 2 attention heads, and a single linear decoder layer. For IMDB the sequence length is 512, for Shakespeare 80. For Shakespeare rc-non-iid, we follow the non-iid scheme from Leaf (Caldas et al., 2019), such that different plays (total of 25) are distributed over different device groups. The results in Table 2 show that CoCoFL reaches significantly higher final accuracies than the state of the art.

*In summary, CoCoFL reaches higher accuracies in almost all presented scenarios. Additionally, CoCoFL preserves fairness (accuracy parity) by enabling constrained devices to contribute to the global model. We attribute this large accuracy gap w.r.t the baselines to the fact that CoCoFL allows any device to calculate gradients based on the full NN, while still reducing required resources, as opposed to state-of-the-art techniques that calculate gradients on subsets of the filters of the NN.*

### 6.3 Ablation Study

We conduct an ablation study to quantify the gains and the error of quantization and operator fusion of frozen blocks in CoCoFL. For this purpose, we modify the MobileNet/CIFAR10 iid experiment of Section 6.1. Instead of three, we have two groups: 10% *strong* devices, i.e., no constraints (training the full NN). We label the remaining 90% *limited* devices, with a computation and memory limit $l$, s.t. $t_{\text{limited}}(A) = \frac{1}{l} \cdot t_{\text{strong}}(A)$ and $M_{\text{limited}} = l \cdot M_{\text{strong}}$. We apply no communication constraint, therefore, $S_{\text{limited}} = S_{\text{strong}}$. Consequently, a *limited* device has to select a configuration $A$ that satisfies both the computation and the memory constraint. Several experiments are conducted, where $l$ is varied between $l \in [0, 1]$. The remaining hyperparameters are kept the same (Table 1). We introduce three variants of CoCoFL, where we profile each variant's configurations $A$ to measure the execution time and the memory footprint:

- CoCoFL$^{\text{F}}$, where only freezing and no quantization or operator fusion is applied. The set of feasible configurations is denoted as $\hat{\mathcal{A}}^{\text{F}}$.
- CoCoFL$^{\text{FF}}$ is a variant where freezing and fusion of operators, but no quantization is applied. Configurations are denoted as $\hat{\mathcal{A}}^{\text{FF}}$.
- CoCoFL$^{\text{QFF}}$ is the mainline variant (Section 6.1). The configurations are equivalent to $\hat{\mathcal{A}}$.

**Quantification of the Error:** To quantify the error introduced through fusion and estimation of the statistics as well as quantization noise, we run all three variants CoCoFL$^{\text{F}}$, CoCoFL$^{\text{FF}}$, and CoCoFL$^{\text{QFF}}$ for different values of $l$, but each variant uses the configurations in $\hat{\mathcal{A}}^{\text{F}}$ on the *limited* devices (ignoring all computation/memory gains that come from quantization and fusion). For a given $l$, all variants can train exactly the same configurations, hence, the same number of blocks. This allows studying the introduced errors independently of the gains in performance/memory. On the right part of Fig. 5a, the cumulative number of trainable blocks for a given $l$ in $\hat{\mathcal{A}}^{\text{F}}$ is displayed. Using $\hat{\mathcal{A}}^{\text{F}}$, at least $l = 0.35$ is required to train a single block. The accuracy results are visualized in Fig. 5a (left), where the accuracy for different values of $l$ is reported on the left. Overall, the error is mostly below 1 p.p., with a maximum of 2.3 p.p. for CoCoFL$^{\text{QFF}}$ and 1.7 p.p. for CoCoFL$^{\text{FF}}$. Note that this analysis ignores the performance and memory gains, which are studied in the next section.

**Quantification of the Gains:** To quantify the gains, we run the experiments with all three variants where each variant uses its own profiling results. Hence, CoCoFL$^{\text{QFF}}$ and CoCoFL$^{\text{FF}}$ use $\hat{\mathcal{A}}$, and $\hat{\mathcal{A}}^{\text{FF}}$, respectively. We measure that CoCoFL$^{\text{QFF}}$ reaches a maximum reduction of 75% of computation time and 60% of memory w.r.t. to CoCoFL$^{\text{F}}$ (45% and 28% with CoCoFL$^{\text{FF}}$). Therefore, at a given constraint $l$, CoCoFL$^{\text{QFF}}$ and CoCoFL$^{\text{FF}}$ can train more configurations and hence, more importantly, configurations with more trained blocks. This results in an overall higher accuracy in FL as can be seen in Fig. 5b (left), where with the same constraint of $l = 0.25$ CoCoFL$^{\text{QFF}}$ achieves an increase of the final accuracy of 6.5 p.p. over CoCoFL$^{\text{FF}}$, while CoCoFL$^{\text{F}}$ has no configuration on the *limited* devices that satisfies the constraint. At a constraint of $l = 0.4$,

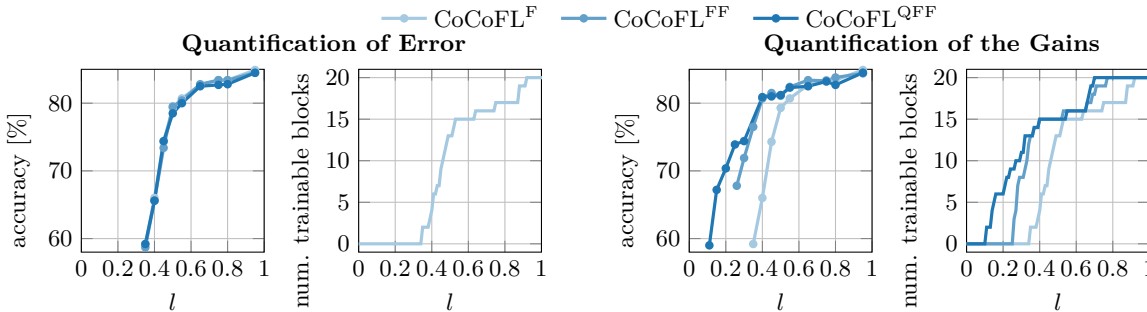

(a) (Left) final accuracy not utilizing gains (error analysis), (right) cumulative number of trainable blocks used for training. When ignoring the gains, the introduced errors from quantization and fusion are negligible.

(b) (Left) final accuracy utilizing the gains, (right) cumulative number of trainable blocks used for training. When utilizing the gains, *limited* devices can train more blocks and achieve higher accuracies, outweighing the error.

Figure 5: Quantification of the gains and the error.

CoCoFL$^{\text{QFF}}$ achieves a final accuracy increase of 14.9 p.p. over CoCoFL$^{\text{F}}$, while CoCoFL$^{\text{FF}}$ achieves a final accuracy increase of 14.3 p.p. over CoCoFL$^{\text{F}}$. From the results it can be concluded that the more blocks the *limited* devices can train, the higher the final accuracy. This is visualized in Fig. 5b (right), where cumulatively the total number of blocks that can be trained for a given constraint is plotted. The figures also show that in the case of the constraint approaching $l = 1.0$, the advantage of quantization and fusion is vanishing, and can even result in small accuracy losses due to the introduced error.

*In summary, for limited devices, the benefits of fusion and quantization of blocks, i.e., training more blocks with the same available resources, largely outweigh the introduced error. Only when the devices' constraint approaches full resources, the gains do vanish. Overall, quantization and fusion increase the FL system's accuracy, as limited devices can make a higher contribution to the model.*

## 7 Conclusion

We proposed CoCoFL that is able to better incorporate knowledge from constrained devices into the FL model, especially in non-iid cases, preserving fairness among participants. Our comparison with the state of the art, based on real hardware measurements, shows that CoCoFL reaches significantly higher final accuracies. We believe that the gains through quantization can even be higher on devices like smartphones that have on-chip integer NN accelerators.

In an FL system, devices are acquiring data through sensing or interaction with the environment. As devices are distributed in the system, they may have access to different types of data. Examples include sensors that sample environments that differ from each other, or smartphones that interact with users with different behaviors. This is therefore important to guarantee that we learn from all these devices, regardless of their capabilities, as any piece of the gathered data matters. What is then important is to provide fairness (accuracy parity) among devices, fairness of participation alone, as was the focus of state of the art, is not enough. By approaching accuracy parity among devices, CoCoFL makes FL systems applicable to a broader range of use cases, especially use cases when the distribution of classes across devices is skewed.

### Acknowledgments

This work is in parts funded by the Deutsches Bundesministerium für Bildung und Forschung (BMBF, Federal Ministry of Education and Research in Germany). The authors acknowledge support by the state of Baden-Württemberg through bwHPC.

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

## A Profiling Setup Details

**Profiling**: Profiling of the used NNs is done for CoCoFL as well as HeteroFL and FjORD to quantify the reduction in training time, maximum memory usage, and upload volume. Two platforms are used: An x86-64 AMD Ryzen 7 with 64 GB RAM and a Raspberry Pi 4 (Cortex-A72 ARM v8 64-bit) with 8 GB RAM. The profiling is done for all tested NN architectures to acquire all configurations in $\hat{\mathcal{A}}$. Similarly, we profile HeteroFL and FjORD where we vary the drop ratio from $0.1 - 1.0$ with 50 linearly spaced steps. We allow freezing of the last layer (linear layer) $b_N$ but do not apply quantization. Profiling takes (MobileNet) 17 min on x64 and about 1 h on ARM. The profiling procedure measures the following quantities in the training:

- Maximum memory: The following training-related parts are included in the memory measurements: First, the model is loaded from disk, second, a training batch is loaded from the disk, and third, an optimizer is initiated. These operations are followed by training of 16 mini-batches. The maximum memory is measured by using the Linux syscall `getrusage()`, with parameter `RUSAGE_SELF`. This call returns a struct with a variable `ru_maxrss`, that stores the maximum amount of memory, the process required at some point in time. For all measurements, we subtract the maximum memory before the training (e.g., the overhead of the process, PyTorch, and NumPy imports).
- Training time: For the same training procedure, we measure the training time. The time measurements include the NN's forward pass execution, setting gradients to zero, the backpropagation step, and optimizer steps. We do not account for mini-batch-level switching between configurations for FjORD and model the switching as a zero-overhead operation.
- Upload volume: The upload volume can be directly calculated from the size of the NNs' `state_dicts` by filtering for parameters that required gradient calculations during training.

**SOTA Comparison:** For HeteroFL and FjORD we enforce the same constraints from Eq. (2) as for CoCoFL. To ensure this we configure the baselines in the following way:

- FjORD (Horvath et al., 2021): In FjORD all devices switch between different configurations for each mini-batch. We use FjORD's drop levels of $\mathcal{P}^{\text{FjORD}} = \{20\%, 40\%, \dots, 100\%\}$. A device $c$ with lower resources can only train with levels $p_j$, that satisfy the constraint $\mathcal{P}_c^{\text{FjORD}} = \{p_j \,|\, t_c(p_j) \leq T \wedge m_c(p_j) \leq M_c \wedge S(p_j) \leq S_c\}$.
- HeteroFL (Diao et al., 2020): Similarly, in HeteroFL the ratio of dropped filters in an NN is set through a shrinkage ratio $k \in (0, 1]$, where $k = 1$ results in training of all available filters. Contrary to FjORD, a device uses the same shrinkage ratio throughout the round. For each device $c$, we set the shrinkage ratio to the maximum value that satisfies the resource constraints $k_c = \max_{0 < k \leq 1} \text{ s.t. } t_c(k) \leq T \wedge m_c(k) \leq M_c \wedge S(k) \leq S_c$.

## B Implementation Details and Hyperparameters

**Miscellaneous Hyperparameters for Vision Models**: We evaluate our technique in an FL setup where we train the NN models DenseNet40 (Huang et al., 2017), MobileNetV2 (Howard et al., 2017), ResNet50 (Gao et al., 2020), and ResNet18 (Gao et al., 2020). The image datasets CIFAR10 (Krizhevsky & Hinton, 2009), CINIC10 (Darlow et al., 2018), CIFAR100 (Krizhevsky & Hinton, 2009), XChest (Wang et al., 2017) and FEMNIST (Cohen et al., 2017) are used, where each individual image has a resolution of $32 \times 32$ pixels, with 3 color channels. In the case of FEMNIST, we scale the $28 \times 28$ grayscale image to $32 \times 32$ and 3 channels to have the same NN structure, independent of the dataset type. Additionally, we do not split the written digits and numbers by writers, as proposed by Caldas et al. (2019), but randomly distribute the images to the devices in case of iid to have an equal amount of data on each device. A mini-batch size of 32 is used for all experiments. For XChest (Wang et al., 2017) we sample 12.7K samples from the full available dataset and train for finding/no finding. The images are downscaled to $256 \times 256$ with 3 color channels. Per round, each active device $c \in \mathcal{C}^{(r)}$ trains for one local epoch. We apply no data augmentation techniques.

**Miscellaneous Hyperparameters for NLP Models:** We evaluate our technique in an FL setup using Transformers (Vaswani et al., 2017) with two datasets. In the case of IMDB (Maas et al., 2011) a *sentence-piece* tokenizer is used with a vocabulary size of $16{,}000$ to detect if a movie review is positive or negative. In the case of Shakespeare (Caldas et al., 2019) every character of the alphabet represents a possible to-

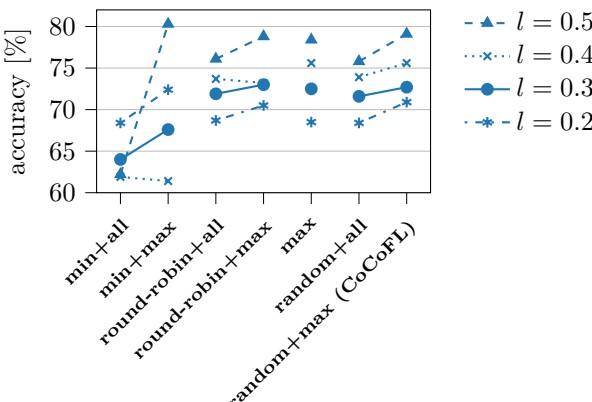

Figure 6: Configuration selection heuristic ablation study evaluating with $l \in [0.2, 0.3, 0.4, 0.5]$.

ken. For Transformer models the baselines HeteroFL and FjORD scale down the feature embedding and hidden size instead of training with subsets of CNN filters. To adjust to the resource requirements we use $[12.5\%, 40\%, 62.5\%, 81.25\%, 100\%]$ of the hidden/embedding dim, yet, for IMDB a large memory overhead remains, hence, *weak* devices have to be dropped from the training.

**Centralized Experiments**: For centralized experiments, we reduce the number of rounds $R$ by a factor of 10, hence, per experiment, we train for $\frac{R}{10}$ epochs over the full dataset. We adjust the learning rate decay steps accordingly.

## C  Configuration Selection Ablation Study

To verify the robustness of our configuration selection heuristic (Section 5), specifically, the per-device per-round random selection of a configuration out of $\mathcal{A}_{\max}$, we perform several experiments. We reuse the setting from the ablation study (Section 6.3) using MobileNet with CIFAR10, 10% *strong* devices and 90% *limited* devices. The *strong* devices train the NN end-to-end. We run several experiments with $l \in [0.2, 0.3, 0.4, 0.5]$ to verify that our heuristic is robust within a large range of constraints (and available configurations). Firstly, we study the effect of our configuration reduction mechanism. Specifically, we compare:

- **max:** Keeping only maximal configurations $\mathcal{A}_{\max} \subseteq \mathcal{A}_f$, i.e., configurations that are not a subset of other feasible configurations (as used in CoCoFL).
- **all:** Keeping all feasible configurations $\mathcal{A}_f \subseteq \hat{\mathcal{A}}$.

We further compare our random approach against other mentioned baselines, such as

- **max**: Using the configuration that trains the maximum number of blocks within the device's capabilities. The combination of this selection mechanism and both reduction mechanisms (**max+max** and **max+all**) result in the same configurations selected, hence, we only evaluate it once.
- **min**: Training the configuration with the minimum number of blocks.
- **round-robin**: Switching between feasible configurations in a round-based manner (all limited devices train the same configuration in a round).
- **random**: Randomly switching between configurations (as used in CoCoFL).

We provide the average final accuracy of three independent runs after 1000 rounds in Fig. 6. We observe that in almost all cases, using only maximal configurations (i.e., configurations that are not a subset of another feasible configuration) increases the final accuracy independent of the selection strategy. Further, we observe that **random+max** (as done in CoCoFL), independent of $l$, is within the highest-performing selection strategies. It can be observed from the results that training the same blocks on all devices does not improve upon random, as **round-robin+max** performs worse than **random+max**. Depending on $l$, **max** and **min+max** can outperform **random+max**, but not consistently throughout different values of $l$.

