# OpenReview forum: "CoCoFL: Communication- and Computation-Aware Federated Learning via Partial NN Freezing and Quantization"
_TMLR — Accepted by TMLR_

### Review · Reviewer_pD8x · 2023-03-30

**Summary Of Contributions:**

The paper proposes a new federated learning technique called CoCoFL that adapts to per-device communication, computation, and memory constraints while still effectively learning from all data on all devices. The approach combines partial freezing and quantization to reduce training complexity and enable devices to select trained/frozen blocks based on their available resources to maximize accuracy under constrained resources. The paper presents a heuristic configuration selection algorithm and an aggregation method for partial updates. The approach is evaluated on various datasets and compared to existing FL methods, showing improved accuracy and reduced training time and communication costs. The contributions of the paper include the development of a novel approach for resource-constrained federated learning, the demonstration of improved accuracy and efficiency compared to existing FL methods, and the proposal of a heuristic configuration selection algorithm and an aggregation method for partial updates.

**Audience:**

Yes

**Broader Impact Concerns:**

no obvious concerns on broader impact


**Claims And Evidence:**

Yes

**Requested Changes:**

1. Discussions on whether the int8 is the best choices, as we know the weights can be typically quantized to 4 bits; and also why not consider low bitwidth floating point.
2. For some devices, the processing unit/chip of float32 and int8 are different whether the proposed method will increase the communication overhead and degrade performance.

**Strengths And Weaknesses:**

Strengths
1. Proposes a novel approach called CoCoFL for resource-constrained federated learning that adapts to per-device communication, computation, and memory constraints while still effectively learning from all data on all devices.

2. Combines partial freezing and quantization to reduce training complexity and enable devices to select trained/frozen blocks based on their available resources to maximize accuracy under constrained resources.

3. Outperforms existing FL methods in terms of accuracy and efficiency and achieves fairness (accuracy parity) by enabling constrained devices to contribute to the global model.

Weaknesses
1. Why int8 is the best choice? More discussions on FP16, FP8, Brain float should be added.

---

> ### Author Response · Authors · 2023-04-07
> **Response to Reviewer pD8x**
>
> We thank the reviewer for the review of our manuscript.
>
> ### Quantization Level (int8, int4, bfloat)
> We use int8 in CoCoFL since it is currently the only quantization level that is fully supported by PyTorch, i.e., all required operators are available. Specifically, we use the FBGEMM quantization backend on x64 and the QNNPACK backend on ARM.
> We expect that other quantization levels have an impact on accuracy, but at the same time, they could allow for a different number of layers trained by a less capable device. The speed improvements through quantization are hard- and software dependent. To have a fair comparison with the state of the art we benchmark CoCoFL on real hardware and all accuracy improvements are based on real hardware measurements. Therefore, we are currently limited use int8. We see no limitation why CoCoFL could not run with other quantization levels as soon as supported by PyTorch and run efficiently on the hardware.
> We would add a respective section in the revised paper to highlight these arguments.
>
> ### On-Chip Communication Overhead
> We presume that this question targets on-chip communication between int8 and float32 units. Such overheads are highly architecture-specific. Our experiments are performed on CPU-based systems (a Raspberry Pi with an ARM SoC and a Ryzen 7 x64 CPU). While there are dedicated FPU and integer units within the CPU, they operate on the same caches and main memory. Therefore, additional on-chip communication due to switching between float32 and int8 is negligible. Also, by performing experiments on real hardware, we fully consider all overheads. Similar behavior would be expected when using GPUs. For instance, CUDA cores combine FP and INT units that operate on the same memory. Other architectures that use dedicated off-CPU NN accelerators with dedicated memory, e.g., scratchpad memory, would likely show higher on-chip communication overheads, as there may be no cache coherence and switching between the CPU and the accelerator would require cache flushes/invalidations. We nevertheless expect that these overheads are rather small (CoCoFL switches at most twice between fp32 and int8 by only training a single coherent block), but further investigation would be required to fully characterize these overheads for such additional architectures. This is beyond scope of our work, as PyTorch does not fully support the required quantized operators for CoCoFL on GPU backends or even on dedicated accelerators.

---

> > ### Comment · Reviewer_pD8x · 2023-05-28
> > **Thanks for the response**
> >
> > I appreciate the responses from the authors on the quantization level and communication overhead. They largely address my concerns.

---

### Review · Reviewer_zCQ5 · 2023-04-03

**Summary Of Contributions:**

The authors propose a method to train a model through federated learning, specific to the situation where client devices are heterogeneous wrt. computing power and upload capacity. Their method, CoCoFL proposes a method which ensures that all devices return an update within a fixed time-window. Given the heterogeneous landscape, this is achieved by selecting specific subsets of the NN to train, while keeping the rest frozen & quantized. The frozen network components need not be communicated, therefore saving bandwidth. The authors propose a heuristic for selecting which network-components to freeze.

Empirically, the authors show that by enforcing $some$ update by each client within the given time-constraint, the weaker devices' data is represented better in the final model than with alternative methods.

**Audience:**

Yes

**Broader Impact Concerns:**

No concerns.

**Claims And Evidence:**

Yes

**Requested Changes:**

I would encourage the authors to consider:

- BN discussion: The authors use and discuss BatchNorm. Specifically, they discuss "Fusion". While not stating it explicitly, it could be misunderstood that this is a contribution to the field - I would encourage the authors to either reformulate or add a citation. The authors claim that "In summary ... reducing the number of MACs". While fusion does reduce the number of operations, it does not change the number of MACs (BN is a scalar operation at test-time). Finally, the authors detail an equation for $y_{iCL}$, for the convolution, which includes a bias-term $b_i$. If followed by BN, adding a learnable bias-term to the linear layer is not necessary, as any bias is captured through $\beta_i$ of BN. These requested change are not critical to securing my recommendation for acceptance.
- BN in FL: For FL specifically, the use of BatchNorm is somewhat critical. Specifically (as in your experiments), non-iid data across the federation breaks the BN-assumption that mini-batch statistics represent the global statistics. There is empirical evidence that BN with non-iid minibatches doesn't work so well in general (https://arxiv.org/abs/1702.03275) and leads generally to worse performance than alternatives such as e.g. GroupNorm or LayerNorm. Furthermore, afaik it is reasonable to assume that transferring local BN-statistics to the server leads to worse privacy. I would ask the authors to clarify how the server computes the global BN-statistics in their experiments. Replacing BN with alternative operations would strengthen their experimental results and be more closely aligned with common practice in the field.
- Exposition of quantized computations. I do not understand why the inputs to quantized operators are re-scaled on-the-fly while $s_o$ is transferred from the server. Assuming BN is fused, the input to a frozen layer \bar{x}\_{l-1} is equal to $ quant(f(o_{l-1})) $, i.e the output of the previous layer passed through an activation function and then quantized. How does the activation function influence the fact that $o_{l-1}$ needs to be specially treated as opposed to on-the-fly quantization of $f(o_{l-1_)$ as the input to the following layer? Similarly, why is the backwards-scale transmitted from the server? How does the server procure these scales? I feel like I'm missing something here and I would like the authors to provide more detail in order to secure my recommendation for acceptance.
- In Figure 2 (right hand side), CoCoFL requires a higher memory-footprint than HeteroFL and FjORD for a given normalized training time. Why is that? (Not critical)
- I believe the first claim you make about CoCoFL on page 2 to be somewhat misleading: "In this paper, we propose a new technique CoCoFL (Fig. 1) that allows all devices to train the full model, irrespective of their capabilities, [...]". I would argue that freezing parts of the model makes it such that a limited-resource device does exactly not train the full model. More nuance would be appropriate, such as for example; "in expectation". (Not critical)

**Strengths And Weaknesses:**

Strengths:
- Paper clarity, writing and exposition. The paper is easy to follow and a joy to read
- Convincing experimental results
- Real-world measurements taken
- Code provided

Weaknesses:
- Novelty: The paper is an example for a well-executed study of a combination of existing ideas in the space. From a critical perspective, the major novelty lies in the heuristic for selecting sub-networks to train. TMLR does not require novelty per se, so I believe this is not a concern for publication. There is sufficient and insightful knowledge contribution. I certainly learned something.
- Some details missing around BN and quantization

---

> ### Author Response · Authors · 2023-04-14
> **Response to Reviewer  zCQ5 (BN in FL)**
>
> We would like to thank the reviewer for the detailed and fair review of our manuscript.
>
> ## Alternatives to BatchNorm in CNNs for FL
>
> We use the state-of-the-art CNNs ResNet/DenseNet/MobileNet with BatchNorm as they are also implemented in the baselines HeteroFL and FjORD and we aim to directly compare against them.
>
>   * The use of BatchNorm allows for fusion, which increases the speed-ups gained in training but is not necessarily required. For example, we provide results for Transformers which do not use BatchNorm but LayerNorm. Our results show, that also in these cases CoCoFL significantly outperforms the state of the art.
>
>   * To show that CoCoFL also works with other normalization techniques in CNNs, we repeated the MobileNet experiment with CIFAR10 with GroupNorm. Our profiling shows that with GroupNorm instead of BatchNorm, CoCoFL loses some improvements w.r.t computation, while memory and communication stay the same.  The results show that also with GroupNorm, CoCoFL's properties of a higher fairness w.r.t. group accuracy remain and the general trends of the algorithms stay the same. **We provide Tables below and attach the GroupNorm version of Figure 4b in the supplementary material**. CoCoFL outperforms the baselines in all scenarios. We observe that with GroupNorm, all algorithms require more rounds to converge, but in non-iid cases, the accuracy over rounds is more stable. We will add a short discussion about BatchNorm and add the GroupNorm results in a revised version of the manuscript.
>
> In the case an NN uses BatchNorm, we apply fusion. Consequently, BatchNorm statistics are required. In CoCoFL, we upload the statistics of the trained layers to the server and average them alongside the NN's parameters. Devices apply fusion using the averaged statistics from the server. We will provide more details about BatchNorm handling in the revised version of the manuscript.
>
> ### Cifar10 MobileNet (groupNorm) experiments
>
> |   | - (iid)  | non-iid@0.1  | rc@0.1  |
> |---|---|---|---|
> | *FedAvg* (full res.) | 76.1 $\pm$ 0.1  | 69.4 $\pm$ 0.5  | 72.9 $\pm$ 1.4  |
> | *CoCoFL* (ours)  | **71.3** $\pm$ **0.1**  |  **61.2** $\pm$ **1.4** |  **63.6**  $\pm$ **3.5**  |
> | *FjORD*  | 64.4 $\pm$ 0.6  | 42.9 $\pm$ 1.7  | 47.0 $\pm$ 5.0  |
> | *HeteroFL*  | 64.8 $\pm$ 0.1  | 55.2 $\pm$ 0.3  | 47.5 $\pm$ 5.0  |
> | *FedAvg*  | 56.2 $\pm$ 0.8  | 52.8 $\pm$ 1.1  | 45.9 $\pm$ 4.7  |
>
>
> ### Cifar10 MobileNet (groupNorm) experiments (fairness evaluation)
> Data is resource correlated. The table shows GroupNorm results for the same setting as visualized in Figure 4b in the manuscript.
>
> |   | combined  | group (strong)  | group (medium)  | group (weak) |
> |---|---|---|---|---|
> |*FedAvg* (full res.) |72.9 $\pm$ 1.4 |75.6 $\pm$ 1.5 |70.8 $\pm$ 4.4 |72.3 $\pm$ 1.6 |
> |*CoCoFL* (ours) |63.7 $\pm$ 3.5 |72.1 $\pm$ 4.5 |60.7 $\pm$ 5.6 |58.3 $\pm$ 9.8 |
> |*FjORD* |47.3 $\pm$ 5.0 |80.7 $\pm$ 7.2 |42.8 $\pm$ 8.5 |18.3 $\pm$ 14.4 |
> |*HeteroFL*  |47.7 $\pm$ 5.0 |81.3 $\pm$ 7.1 |42.8 $\pm$ 9.0 |19.0 $\pm$ 13.5 |
> |*FedAvg* |46.2 $\pm$ 4.7 |78.7 $\pm$ 7.4 |42.1 $\pm$ 8.1 |17.7 $\pm$ 13.2 |

---

> ### Author Response · Authors · 2023-04-14
> **Response to Reviewer zCQ5 (Exposition of quantized computations)**
>
> ## Details of Quantization in CoCoFL/PyTorch
>
> We would like to point out that the handling of the scales for quantization is not strictly a conceptual part of CoCoFL, but a necessity that comes with an efficient implementation in PyTorch. We observe that the scales stabilize quickly and only slowly change over the rounds.
>
>   * **Input scale**: We calculate the input scale $s_x$ on-the-fly on the device as we discover that the overhead is negligible since the input exists in float representation anyway. For example, in the forward pass, this has to be done only once for all type (a) blocks.
>   * **Output Scale**: Output scales $s_o$ are required for all operations where the quantization backend (FBGEMM and QNNPACK) utilizes int16/32 accumulation (convolutions, matrix multiplication, add operations, but not activation functions like ReLU). Each operator takes quantized tensors (e.g., weights and activations) as input and requires an output scale $s_o$, as they return int8 and the backends do not implement direct access to the int16/32 tensor. This is an limitation of the PyTorch backend that does not allow for efficient on-the-fly scale calculation. Consequently, to calculate the output scale, the output in float has to be known. On-the-fly calculation would require performing the whole operation twice, first in float (e.g., the fused convolution or linear layers in Transformers) to apply Equation 5 on the float output (calculation of $s_o$) and then again quantized with the respective output scale set. Performing the whole operation in float would have a significant impact on CoCoFL's computation speed and memory gains. The same is true for the output scale of the backwards operation. To calculate the scale on-the-fly, it would be required to calculate the gradients first in float to calculate the scale. Therefore, these scales are acquired from the server: Devices that train a respective layer, do calculate the operation in float anyways, hence, can with low overhead calculate the output's scale and transmit it to the server for the next FL round. Similarly, for the backwards output scale, we apply Equation 5 on the float gradient to procure the scales. This can be even done only once in the last batch. The overheads are minimal and fully considered in our profiling. Together with the NN's parameters, the scales are averaged on the server.
>   * **Activation Function**: Activation functions like ReLU do not utilize int16/int32 accumulation in PyTorch, therefore, it is not required to rescale the activation function's output. Consequently, the output has the same scale as the input.
>
> We hope that our additional explanations help to understand our implementation in PyTorch. We will add the justification of partly using on-the-fly scale calculations and scales from the server in the revised version of the manuscript. Alongside the implementation details in the paper, we provide the code for further details and full reproducibility.

---

> ### Author Response · Authors · 2023-04-14
> **Response to Reviewer zCQ5 (BN discussion / Figure 2 / Claim)**
>
> ## BatchNorm Discussion
> We see that our explanation gives the impression that BatchNorm fusion is a contribution of CoCoFL. We will add appropriate citations in the revised version of the manuscript. We will also change the formulation from "[...] reducing the number of MACs" to "[...] reducing the number of operations". When BatchNorm layers follow convolutional layers, the convolutional layer's bias is usually deactivated. We will simplify our equation in the revised version of the manuscript to reflect the commonly used implementation without the convolutional layer's bias.
>
> ## Figure (Training Time vs. Memory)
> We presume that this question targets Figure 3 and not Figure 2.
> Figure 3 (right) shows the possible *configurations* for each algorithm and that CoCoFL supports the same range w.r.t. to the constraints as SOTA, but CoCoFL enables independent adjustability of computation/memory and communication (Figure 3 left). Since SOTA and CoCoFL use different mechanisms, the trade-offs between communication/computation/memory vary:
>
>   * In HeteroFL/FjORD, a *configuration* refers to a specific ratio of filters that are trained and dropped filters.
>   * In CoCoFL, a *configuration* refers to a certain selection of blocks that are frozen/quant./fused and blocks that are trained. For some cases (as pointed out by the reviewer), this means CoCoFL has to pick a non-optimal configuration w.r.t. computation to satisfy the memory constraint, wasting some computational resources. However, in our results, we show that this is still more effective w.r.t. accuracy than using subsets as used by the state of the art.
> We will add these details to the description of Figure 3 in the revised version of the manuscript.
>
> ## CoCoFL Claim
> We see that our formulation may give the impression that CoCoFL allows less capable devices to train the NN *end-to-end*, which is not the case. Rather, CoCoFL allows less capable devices to calculate *some* gradients of the NN layers based on all parameters (which is different to subset-based technqiues). We will update this description in the revised version of the manuscript.

---

> > ### Comment · Reviewer_zCQ5 · 2023-06-09
> > **Thanks for the revision**
> >
> > I thank the authors for the additional explanations and clarifications. They address my concerns. Very interesting work.

---

### Review · Reviewer_mk4Z · 2023-05-18

**Summary Of Contributions:**

The authors study the Federated Learning (FL) problem in a synchronous setting where the client needs to send an update to a central server within a fixed time constraint per round. Under this setting, the authors present CoCoFL which allows edge devices with limited computational resources to train full neural networks. Along these lines the main contributions are:

1) CoCoFL allows the client devices to train on the full neural network architecture by using only a subset of trainable layers during training. This is achieved by a combination of freezing, fusing and quantization of a subset of neural network blocks to reduce computational overhead. The rest of the blocks are trained at full precision.

2) The authors rely on profiling to select plausible neural network configurations. To prevent combinatorial explosion, the authors propose a heuristic to select configurations which allow training contiguous range of blocks and then selecting a random configuration from the resulting set.

3) The authors provide empirical justification for the method using neural network backbones like DenseNet, MobileNet, ResNet18 etc.


**Audience:**

Yes

**Claims And Evidence:**

Yes

**Requested Changes:**

Requested Changes:

1) The authors select a “block” as the most granular unit on which freezing and quantization are applied. However, a more granular unit might be a single neural layer. Is there a particular reason for selecting the former in favor of the latter apart from combinatorial complexity during configuration selection? If so, please include in subsequent revisions.

2) Though the proposed heuristic (selection of training contiguous blocks) for configuration selection works well empirically, is there a particular reason for the choice of heuristic. If the authors tried some other heuristics which did not perform as well, it might be worth including a small ablation experiment (in the main text or appendix) regarding the same on a couple datasets.

3) The authors discuss layer fusion in context of BatchNorm in Section 4.2. However, its fairly common to use variants like LayerNorm or GroupNorm. Moreover, Can the authors include the relevant analysis in subsequent revision.

4) The paper is well-written and easy to follow. However, it might be better to discuss the fusion and implementation issues for attention blocks in language models in Section 4. The authors currently briefly state some of this in Section 6.2


**Strengths And Weaknesses:**

Strengths:

1) The motivation behind the proposed methodology is clear to me and makes sense. The proposed heuristics for configuration selection and the idea of freezing layers instead of removing them (as in subset selection methods) is novel.

2) The authors show strong empirical evidence across a number of datasets and neural net backbones in Table 2 over competing baselines.

Weaknesses:
See Requested Changes

---

> ### Author Response · Authors · 2023-05-26
> **Response to  Reviewer mk4Z (Granularity/Heuristic)**
>
> We would like to thank the reviewer for the detailed and fair review of our manuscript.
>
> ## Block Granularity in CoCoFL
> For CNNs, we define a *block* as a single Convolutional Layer (CL) + normalization (+ReLU) as the smallest entity that our technique allows to either train or freeze (a block in this context should not be confused with a *DenseNet/ResNet bottleneck* block or similar). In the experimental setup (Section 6.1), for ResNet, we have as many blocks as there are *ResNet bottleneck blocks* (similar to DenseNet, MobileNet, and Transformers). The reason that we select this granularity of blocks is that they always have the *same* structure, hence, lowering the PyTorch implementation overhead. In theory, CoCoFL could also train ResNet with only a single CL layer as a block, but that would increase the implementation complexity, as some parts of the ResNet block are frozen while others are trained, requiring many cases to be implemented seperately. While it would further increase the granularity, we do not expect an significant effect on accuracy. Additionally, the supported resource range of CoCoFL would not significantly increase. The implementation complexity in PyTorch would increase even more in the case of transformers, as there are 6 linear layers in an EncoderBlock that can be individually frozen or trained. We will add this argument in the revised version of the manuscript.
>
>
> ## Block Selection Heuristic Ablation Study
> To answer this question, we perform several experiments to verify the robustness of our heuristics in various resource scenarios. We reuse the setting from the ablation study (Section 6.3) using MobileNet with CIFAR10, $10$\% *strong* devices and $90$\% *limited* devices. The strong devices train the NN end-to-end. We run several experiments with $l \in [0.2, 0.3, 0.4, 0.5]$ to verify that our heuristic is robust within a large range of constraints (and available configurations).
>
> Firstly, we study the effect of our configuration reduction mechanism. Specifically, we compare:
>
>   * **max**: Keeping only maximal configurations $\mathcal{A}_{\text{max}}{\subseteq}\mathcal{A}_f$,i.e., configurations that are not a subset of other feasible configurations (as used in the main paper).
>   * **all**: Keeping all feasible configurations $\mathcal{A}_f{\subseteq}\hat{\mathcal{A}}$.
>
> We further compare our random approach against other mentioned baselines, such as
>
>   * **max:** Using the configuration that trains the maximum number of blocks within the device's capabilities. The combination of this selection mechanism and both reduction mechanisms (**max+max** and **max+all**) result in the same configurations selected, hence we only evaluate it once.
>   * **min:** Training the configuration with the minimum number of blocks.
>   * **round_robin:** Switching between feasible configurations in a round based-manner (all limited devices train the same configuration in a round).
>   * **random:** Randomly switching between configurations (as used in the main paper).
>
> | $l$ |**min+all** | **min+max**  | **round-robin+all**  | **round-robin+max**  |  **max**  |   **random+all**  |   **random+max**  |
> |---|---|---|---|---|---|---|---|
> | 0.5 | 62.3 | 79.8 | 76.0 | 79.3 | 78.0 | 76.0 | 79.3 |
> | 0.4 | 63.3 | 63.1 | 68.6 | 73.8 | 75.2 | 74.5 | 76.7 |
> | 0.3 | 64.6 | 68.9 | 72.1 | 71.9 | 72.5 | 71.8 | 72.5 |
> | 0.2 | 69.8 | 71.1 | 67.5 | 70.0 | 68.0 | 68.1 | 71.1 |
>
> We observe that in almost all cases, using only maximal configurations (i.e., configurations that are not a subset of another feasible configuration) increases the final accuracy independent of the selection strategy. Further, we observe that **random+max** (as done in the main paper) is consistently within the best performing. **round-robin** does not improve upon **random**. Depending on $l$, **max** can be similar to **random**, however, it can also result in lower accuracy, as some blocks do not receive training. Using **min** results in the lowest accuracy, except for $l=0.2$, where **min** outperforms **max**. We will add these additional experiments in the appendix of the revised version of the manuscript.

---

> ### Author Response · Authors · 2023-05-26
> **Response to Reviewer mk4Z (Normalization/Transformer)**
>
> ## Alternatives to BatchNorm in FL
> We would like to thank the reviewer for this comment. This point has also been raised by reviewer zCQ5.
> We use the state-of-the-art CNNs ResNet/DenseNet/MobileNet with BatchNorm as they are also implemented in the baselines HeteroFL and FjORD, and we aim to directly compare against them. However, CoCoFL is not limited to CNNs with BatchNorm:
>
>   * To show that CoCoFL also works with other normalization techniques, we repeated the MobileNet experiment with CIFAR10 with GroupNorm. Our profiling shows that with GroupNorm instead of BatchNorm, CoCoFL loses some improvements w.r.t computation, while memory and communication stay the same. The results show that also with GroupNorm, CoCoFL's properties of a higher fairness w.r.t. group accuracy remain, and the general trend of the algorithms stays the same. (**we provide Tables below and attach the GroupNorm version of Figure 4b in the supplementary material**) CoCoFL outperforms the baselines in all scenarios. We observe that with GroupNorm, all algorithms require more rounds to converge, but in non-iid cases, the accuracy over rounds is more stable.
>
> We will update our manuscript with groupNorm results.
>
> **Cifar10 MobileNet (groupNorm) experiments**
>
> |   | - (iid)  | non-iid@0.1  | rc@0.1  |
> |---|---|---|---|
> | *FedAvg* (full res.) | 76.1 $\pm$ 0.1  | 69.4 $\pm$ 0.5  | 72.9 $\pm$ 1.4  |
> | *CoCoFL* (ours)  | **71.3** $\pm$ **0.1**  |  **61.2** $\pm$ **1.4** |  **63.6**  $\pm$ **3.5**  |
> | *FjORD*  | 64.4 $\pm$ 0.6  | 42.9 $\pm$ 1.7  | 47.0 $\pm$ 5.0  |
> | *HeteroFL*  | 64.8 $\pm$ 0.1  | 55.2 $\pm$ 0.3  | 47.5 $\pm$ 5.0  |
> | *FedAvg*  | 56.2 $\pm$ 0.8  | 52.8 $\pm$ 1.1  | 45.9 $\pm$ 4.7  |
>
>
> **Cifar10 MobileNet (groupNorm) experiments Group Accuracy**
> Data is resource correlated
>
> |   | average  | group (strong)  | group (medium)  | group (weak) |
> |---|---|---|---|---|
> |*FedAvg* (full res.) |72.9 $\pm$ 1.4 |75.6 $\pm$ 1.5 |70.8 $\pm$ 4.4 |72.3 $\pm$ 1.6 |
> |*CoCoFL* (ours) |63.7 $\pm$ 3.5 |72.1 $\pm$ 4.5 |60.7 $\pm$ 5.6 |58.3 $\pm$ 9.8 |
> |*FjORD* |47.3 $\pm$ 5.0 |80.7 $\pm$ 7.2 |42.8 $\pm$ 8.5 |18.3 $\pm$ 14.4 |
> |*HeteroFL*  |47.7 $\pm$ 5.0 |81.3 $\pm$ 7.1 |42.8 $\pm$ 9.0 |19.0 $\pm$ 13.5 |
> |*FedAvg* |46.2 $\pm$ 4.7 |78.7 $\pm$ 7.4 |42.1 $\pm$ 8.1 |17.7 $\pm$ 13.2 |
>
>
>
> ## Placement of Transformer Discussion
> We will move the technical discussion about Transformers from Section 6 to Section 4 in a revised version of the manuscript.

---

### Author Response · Authors · 2023-06-01
**summary of the revision of the manuscript**

We would like to thank all reviewers for their constructive feedback on our manuscript. We have uploaded a revised version of our manuscript. The changes in the text are colored in blue. Specifically, we added/changed:

 * (Reviewer zCQ5) We rephrased the claim made in Section 1.
 * (Reviewer zCQ5) We added citations for BatchNorm Fusion in Section 4 to make clear that BatchNorm fusion for inference is not a contribution of the manuscript.
 * (Reviewer zCQ5) We simplified the BatchNorm Fusion equations to reflect the standard case when no Convolution Layer Bias is used.
 * (Reviewer pD8x) We added arguments why (at the time) int8 is chosen for quantization, as it is the only supported quantization level.
 * (Reviewer zCQ5) We added additional explanations why some scales can be calculated cheaply on-the-fly, while others are gathered from the server.
  * (Reviewer zCQ5) We added details about the handling of BN statistics.
  * (Reviewer mk4Z) We moved the transformer implementation details to Section 4.
  * (Reviewer mk4Z) We added the block granularity discussion to Section 6.
  * (Reviewer zCQ5) We added more explanation of Figure 3 to Section 6.
  * (Reviewer zCQ5 & mk4Z) We added the results of MobileNet with GroupNorm to Table 2. Additionally, we provide explanations of the results in Section 6.2.
   * (Reviewer mk4Z) We added the ablation study w.r.t. to the random selection heuristic to Appendix C (referenced in Section 5).

---

### Decision · Action_Editors · 2023-06-12

**Recommendation:** Accept as is

**Comment:**

The paper addresses an issue in federated learning, where devices have different resources and need to finish training within the same deadline. Existing methods that drop neurons or filters from the neural network on constrained devices lead to inaccuracies. The authors propose a novel federated learning technique that maintains the full neural network structure on all devices by quantizing selected layers to reduce resource requirements while still allowing high accuracy.

Initial concerns include the demand that the paper should provide more explanations on why int8 is chosen as the best option and address potential performance issues when using different processing units. The authors argued they use int8 quantization in CoCoFL because it is fully supported by PyTorch, allowing for accurate measurements on real hardware. They acknowledge that other quantization levels may impact accuracy and enable training on less capable devices. However, PyTorch currently lacks support for these quantized operators on GPUs or dedicated accelerators. Questions on architectural choices were addressed in great detail, and additional ablations were provided.

**Audience:**

Yes.

**Claims And Evidence:**

Yes.